# Fractional Singular Differential Systems of Lane–Emden Type: Existence and Uniqueness of Solutions

**Yazid Gouari [1], Zoubir Dahmani [2], Shan E. Farooq [3] and Farooq Ahmad [4,5,\*]**

[1] Laboratory of Pure and Applied Math's, Faculty of SEI, UMAB University of Mostaganem, Mostaganem 27000, Algeria; gouariyazid@gmail.com

[2] LPAM, UMAB, University of Mostaganem, Mostaganem 27000, Algeria; zzdahmani@yahoo.fr

[3] Mathematics Department, GC University, Lahore 54000, Pakistan; shanefarooq@gmail.com

[4] Mathematics Department, University of Ha'il, Ha'il 55211, Saudi Arabia

[5] School of Mechanical and Aerospace Engineering, NANYANG Technological University, Singapore 637551, Singapore

\* Correspondence: ahmad.farooq@uoh.edu.sa and ahmad.farooq@ntu.edu.sg

**Abstract:** A coupled system of singular fractional differential equations involving Riemann–Liouville integral and Caputo derivative is considered in this paper. The question of existence and uniqueness of solutions is studied using Banach contraction principle. Furthermore, the question of existence of at least one solution is discussed. At the end, an illustrative example is given in details.

**Keywords:** Caputo derivative; lane emden system; existence of solution; singular differential equation

## 1. Introduction

Fractional calculus theory and differential equations of non integer order are new powerful mathematical tools for modeling complex real world phenomena, see for instance the papers [1–3] for more details. We can find many applications for these two theories in mechanics, chemistry, biology, economics, visco-elasticity, electrochemistry, etc. For more information and more details and recent applications, we refer the reader to [4–9]. For the theory of fractional differential equations, in the literature, we can find many authors who have paid much attention to the question of existence and uniqueness of solutions for certain types of fractional equations. For more details, we refer the reader to the papers [10–15]. Moreover, the study of coupled systems of fractional order is also important in various problems of applied sciences, see for instance the two research works [12,16–19].

In this work, a new singular system of differential equations is investigated. The considered problem has some relationship with the well known Lane–Emden equation which has a considerable importance in astrophysics, for more details, one can consult [14,20–22]. We attract the reader's attention that the standard Lane–Emden problem has the form:

$$x'' (t) + \frac{a}{t} x' (t) + f (t, x (t)) = g (t), \, t \in [0,1],$$

with

$$x (0) = A, \, x' (0) = B,$$

and $A$ and $B$ are constants, $f, g$ are continuous real functions.

Let us now recall some other research works that have motivated the present paper. We begin by [23], where Mechee et al. have applied a numerical approach to study the following problem:

$$\begin{cases} D^{\alpha} y (t) + \dfrac{k}{t^{\alpha-\beta}} D^{\beta} y (t) + f (t, y (t)) = g (t), \, t \in [0,1], \\ \\ k \geq 0, \, 1 < \alpha \leq 2, \, 0 < \beta \leq 1, \end{cases}$$

with

$$y (0) = A, \, y' (0) = B,$$

and $A$ and $B$ are constants.

Then, in [20], Rabha W. Ibrahim has studied the question of Ulam Hyers Stability for the following equation:

$$\begin{cases} D^{\beta} \left( D^{\alpha} + \frac{a}{t} \right) u (t) + f (t, u (t)) = g (t), \\ \\ u (0) = \mu, \, u (1) = \nu, \\ \\ 0 < \alpha, \beta \leq 1, \, 0 \leq t \leq 1, \, a \geq 0, \end{cases}$$

where $D^{\gamma}$ is the Caputo derivative, $f$ is a continuous function and $g \in C ([0,1])$.

In [24], Dahmani and Tabia have been concerned with the following general Lane–Emden coupled system of fractional differential equations:

$$\begin{cases} D^{\beta_1} \left( D^{\alpha_1} + \frac{a_1}{t} \right) x_1 (t) + f_1 (t, x_1 (t), x_2 (t), ..., x_n (t)) = g_1 (t), \, t \in J, \\ D^{\beta_2} \left( D^{\alpha_2} + \frac{a_2}{t} \right) x_2 (t) + f_2 (t, x_1 (t), x_2 (t), ..., x_n (t)) = g_2 (t), \, t \in J, \\ \qquad\qquad\qquad\qquad\qquad \vdots \\ D^{\beta_n} \left( D^{\alpha_n} + \frac{a_n}{t} \right) x_n (t) + f_n (t, x_1 (t), x_2 (t), ..., x_n (t)) = g_n (t), \, t \in J, \\ \sum\limits_{k=1}^{n} |x_k (0)| = \sum\limits_{k=1}^{n} \left| x'_k (0) \right| = ... = \sum\limits_{k=1}^{n} \left| x_k^{(l-1)} (0) \right| = 0, \\ \sum\limits_{k=1}^{n} |D^{\alpha_k} x_k (0)| = \sum\limits_{k=1}^{n} |D^{\alpha_k+1} x_k (0)| = ... = \sum\limits_{k=1}^{n} |D^{\alpha_k+l-2} x_k (0)| = 0, \\ D^{\alpha_k+l-1} x_k (1) = 0, \, k = 1, 2, ..., n, \end{cases}$$

where $l - 1 < \alpha_k, \beta_k < l, \, a_k \geq 0, l \in \mathbb{N} - \{0,1\}, \, k = 1, 2, ..., n, \, n \in \mathbb{N} - \{0\}$, and $J := [0,1]$.

The authors have discussed the existence, uniqueness and some types of Ulam stabilities for the proposed coupled nonlinear fractional system.

Recently, A. Bekkouche et al. [25] have studied the existence of solutions and the $\Delta-$Ulam stabilities for the following two dimension non homogeneous Lane–Emden fractional system:

$$\begin{cases} D^{\beta_1}\left(D^{\alpha_1} + b_1 g_1\left(t\right)\right) x_1\left(t\right) + f_1\left(t, x_1\left(t\right), x_2\left(t\right)\right) = \omega_1 S_1\left(t, x_1\left(t\right), x_2\left(t\right)\right), \ 0 < t < 1, \\[2mm] D^{\beta_2}\left(D^{\alpha_2} + b_2 g_2\left(t\right)\right) x_2\left(t\right) + f_2\left(t, x_1\left(t\right), x_2\left(t\right)\right) = \omega_2 S_2\left(t, x_1\left(t\right), x_2\left(t\right)\right), \ 0 < t < 1, \\[2mm] x_k(0) = 0, \ D^{\alpha} x_k(1) + b_k g_k(1) x_k(1) = 0, \end{cases}$$

where $0 < \beta_k < 1, 0 < \alpha_k < 1, b_k \geq 0, 0 < \omega_k < \infty, k = 1, 2$ and the derivatives $D^{\beta_k}$ and $D^{\alpha_k}$ are in the sense of Caputo. The functions $f_k : [0, 1] \times \mathbb{R}^2 \to \mathbb{R}$ and $S_k : [0, 1] \times \mathbb{R}^2 \to \mathbb{R}$ are continuous, $g_k : ]0, 1] \to [0, +\infty)$ is continuous and singular at $t = 0$.

The published papers in [5,17,26,27] have also investigated some singular Lane–Emden type problems.

In [28], Y. Gouari et al. have studied the following nonlinear singular integro-differential equation of Lane–Emden type with nonlocal multi point integral conditions:

$$\begin{cases} D^{\beta}(D^{\alpha} + \dfrac{k}{t^{\lambda}}) y(t) + \Delta_1 f(t, y(t), D^{\delta} y(t)) + \Delta_2 g(t, y(t), I^{\rho} y(t)) + h(t, y(t)) \\ = l(t), \ t \in ]0, 1[, \\ y(0) = 0, \\ y(1) = b \displaystyle\int_0^{\eta} y(s) ds, \ 0 < \eta < 1, \\ I^q y(u) = y(1), \ 0 < u < 1, \\ k > 0, 0 < \lambda \leq 1, \ 1 \leq \beta \leq 2, \ 0 \leq \alpha, \ \delta \leq 1, \end{cases}$$

with the conditions: $\Delta_1 > 0, \Delta_2 > 0, J := [0, 1]$, the derivatives of the problem are in the sense of Caputo, $I^{\rho}$ denotes the Riemann–Liouville integral of order $\rho$, and $f, g$ are two given functions defined on $J \times \mathbb{R}^2$, also $h : J \times \mathbb{R} \to \mathbb{R}$ is a given function and $l$ is given function defined over $J$. The authors have investigated the existence and uniqueness of solutions for the considered class.

Very recently, in [29], the authors have considered the following sequential time-singular fractional problem of Lane–Emden type:

$$\begin{cases} D^{\alpha_1}(D^{\alpha_2}...(D^{\alpha_n}(D^{\beta} + \dfrac{k}{t^{\lambda}}))...) u(t) + f(t, u(t), D^{\delta} u(t)) + g(t, u(t), I^{\rho} u(t)) \\ + h(t, u(t)) = l(t), \ t \in ]0, 1[, \\ u(0) = 0, \\ u(1) = \theta, \\ D^{\alpha_n}(D^{\beta} u(0)) = 0, \\ D^{\alpha_{n-1}}(D^{\alpha_n}(D^{\beta} u(0))) = 0, \\ \vdots \\ D^{\alpha_3}(D^{\alpha_4}...(D^{\alpha_n}(D^{\beta} u(0)))...) = 0, \\ D^{\alpha_2}(D^{\alpha_3}...(D^{\alpha_n}(D^{\beta} u(1) + \phi_{k,\lambda}(1) u(1)))...) = 0, \\ k > 0. \end{cases}$$

where $J := [0, 1], 0 \leq \beta \leq 1, 0 \leq \alpha_i < 1; i = 1, 2, ..., n, \delta < min(\beta, \alpha_i), \phi_{k,\lambda}(t) = \dfrac{k}{t^{\lambda}}$, the sequential derivatives are in the sense of Caputo, $I^{\rho}$ denotes the Riemann–Liouville fractional integral of order $\rho$, and $f, g : J \times \mathbb{R}^2 \to \mathbb{R}$ are two given functions, also $h : J \times \mathbb{R} \to \mathbb{R}$ is a given function

and $l$ is a function which is defined on $J$. The authors of this paper have proved the existence and uniqueness of solutions by application of Banach contraction principle, then, by means of Schaefer fixed point theorem, they have studied the existence of at least one solution for the problem.

Motivated by the above works, in the present paper, we are concerned with the study of a new singular system of Lane–Emden type. In fact, we investigate the following class of singular fractional differential equations with two different orders of derivation and also with a time singularity at the origin:

$$
\begin{cases}
D^{\beta_1}(D^{\alpha_1} + \dfrac{k_1}{t^{\lambda_1}})y_1(t) + f_1(t, y_1(t), D^{\delta_1}y_2(t)) + g_1(t, y_1(t), I^{\rho_1}y_2(t)) + h_1(t, y_1(t)) \\
\quad = l_1(t), \ \ t \in ]0,1[, \\
D^{\beta_2}(D^{\alpha_2} + \dfrac{k_2}{t^{\lambda_2}})y_2(t) + f_2(t, y_1(t), D^{\delta_2}y_2(t)) + g_2(t, y_1(t), I^{\rho_2}y_2(t)) + h_2(t, y_2(t)) \\
\quad = l_2(t), \ \ t \in ]0,1[, \\
y_i(1) = y_i(0), \\
y_i'(1) = y_i'(0), \\
D^{\alpha_i}y_i(1) + \phi_{k_i,\lambda_i}(1)y_i(1) = \displaystyle\int_0^{\mu_i} T_i(\tau)y_i(\tau)d\tau, \\
I^{q_i}y_i(\theta_i) = y_i(1), \\
k_i \geq 0, 1 < \alpha_i, \beta_i \leq 2, 0 < \delta_i, \theta_i, \lambda_i, \mu_i < 1, q_i \geq 0, \rho_i \geq 0, i = 1, 2.
\end{cases}
\tag{1}
$$

The following data are taken into account: $J := [0,1]$, $\phi_{k_i,\lambda_i}(t) = \dfrac{k_i}{t^{\lambda_i}}$, the derivative $D^{\alpha_i}$ is in the sense of Caputo, $I^{\rho_i}$ is the Riemann–Liouville integral of order $\rho_i$, the four functions $f_1, f_2, g_1, g_2$ are defined on $J \times \mathbb{R}^2$, $h_1, h_2$ are defined on $J \times \mathbb{R}$, and $l_1, l_2$ are defined over $J$, and $T_1, T_2$ are continuous over $J$, with $\sup_{t \in J} |T_i(t)| = \chi_i$. $i = 1, 2$.

Regarding the above problem, the reader is invited to take into account the following particular points:

(1.) The Caputo derivative is introduced in both sides of the coupled system.
(2.) Furthermore, the Riemann Liouville integral is introduced in one nonlinearity of the right hand side of each the equation of the considered system.
(3.) Another important point in this paper is the time singularity at the origin for each equation of the above $2D$-system.

These three particular conditions allow us to consider a new fractional system of Lane–Emden type. It is important to note also that Equation (1) is general enough to describe many problems that can arise in mathematical physics since this system includes several particular types of problems that have applications in real word phenomena. For example, one can verify that our problem includes the standard Lane–Emden equation as a special case when $i = 1$. Furthermore, the above system includes the Emden Fowler equation; such an equation has been introduced to model several phenomena in mathematical physics and astrophysics, such as the theory of stellar structure.

To the best of our knowledge, this is the first time in the literature where such problem is investigated.

This paper is structured as follows: we begin by recalling some fractional calculus concepts. Then, by application of the fractional integral inequality theory combined with the fixed point theory, we study the questions of existence and uniqueness of solutions and the existence of at least one solution for the considered singular system. In Section 4, an illustrative example is presented to show the applicability of our main results. Finally, a conclusion follows.

## 2. Preliminaries

We recall some definitions and lemmas that will be used later. For more details, we refer to the reference [30].

**Definition 1.** *Let $\alpha > 0$, and $f : [0,1] \longmapsto \mathbb{R}$ be a continuous function. The Riemann–Liouville integral of order $\alpha$ is defined by:*

$$I^\alpha f(t) = \frac{1}{\Gamma(\alpha)} \int_0^t (t - \tau)^{\alpha-1} f(\tau) d\tau,$$

*where $\Gamma(\alpha) := \int_0^\infty e^{-u} u^{\alpha-1} du$*

**Definition 2.** *For a function $f \in C^n([0,1], \mathbb{R})$ and $n - 1 < \alpha \le n$, the Caputo fractional derivative is defined by:*

$$
\begin{aligned}
D^\alpha f(t) &= I^{n-\alpha} \frac{d^n}{dt^n}(f(t)) \\[2mm]
&= \frac{1}{\Gamma(n - \alpha)} \int_0^t (t - s)^{n-\alpha-1} f^{(n)}(s) ds.
\end{aligned}
$$

Furthermore, we recall the following lemmas [30]:

**Lemma 1.** *Suppose that $n \in \mathbb{N}^*$, and $n - 1 < \alpha < n$. Thus, the general solution of $D^\alpha y(t) = 0$ is:*

$$y(t) = \sum_{i=0}^{n-1} c_i t^i,$$

*such that $c_i \in \mathbb{R}, i = 0, 1, 2, ..., n - 1$.*

**Lemma 2.** *Suppose that $n \in \mathbb{N}^*$, and $n - 1 < \alpha < n$, then, we get:*

$$I^\alpha D^\alpha y(t) = y(t) + \sum_{i=0}^{n-1} c_i t^i,$$

*for some $c_i \in \mathbb{R}, i = 0, 1, 2, ..., n - 1$.*

We prove the following result:

**Lemma 3.** *Suppose that $G_i \in C([0,1])$. Then, the problem:*

$$
\begin{cases}
D^{\beta_i}\left(D^{\alpha_i} + \dfrac{k_i}{t^{\lambda_i}}\right) y_i(t) = G_i(t), \\
\quad y_i(1) = y_i(0), \\
\quad y_i'(1) = y_i'(0), \\
D^{\alpha_i} y_i(1) + \phi_{k_i, \lambda_i}(1) y_i(1) = \displaystyle\int_0^{\mu_i} T_i(\tau) y_i(\tau) d\tau, \\
\quad I^{q_i} y_i(\theta_i) = y_i(1), \\
\quad k_i \ge 0, 1 < \alpha_i, \beta_i \le 2, i = 1, 2,
\end{cases}
\tag{2}
$$

*admits the couple $(y_1(t), y_2(t))$, where*

$$y_i(t) = I^{\alpha_i} R_i(t) + \left( \frac{\theta_i^* \theta_i^{q_i+1}}{\Gamma(q_i+2)} - t \right) I^{\alpha_i} R_i(1) + \left( \frac{t^{\alpha_i}}{1-\alpha_i} - \frac{t^{\alpha_i+1} + \alpha_i t}{(1-\alpha_i)(1+\alpha_i)} - \nu_{1_i} \right) I^{\alpha_i-1} R_i(1)$$

$$- I^{\alpha_i+q_i} R_i(\theta_i) + \left( \frac{t - t^{\alpha_i+1}}{(1-\alpha_i)\Gamma(\alpha_i+1)} + \frac{\alpha_i t^{\alpha_i+1}}{(1-\alpha_i)\Gamma(\alpha_i+2)} - \nu_{2_i} \right) I^{\beta_i} G_i(1) \tag{3}$$

$$+ \left( \frac{t^{\alpha_i} - \alpha_i t^{\alpha_i+1}}{(1-\alpha_i)\Gamma(\alpha_i+1)} - \frac{t}{(1-\alpha_i)\Gamma(\alpha_i+2)} - \nu_{3_i} \right) \int_0^{\mu_i} T_i(\tau) y_i(\tau) d\tau,$$

*as integral solution,*
*where,*

$$R_i(s) = \frac{1}{\Gamma(\beta_i)} \int_0^s (s-\tau)^{\beta_i-1} G_i(\tau) d\tau - \frac{k_i}{s^{\lambda_i}} y_i(s),$$

$$\nu_{1_i} = \phi_i \left( 1 - \frac{\theta_i \Gamma(\alpha_i+1)}{\alpha_i + q_i + 1} \right) - \frac{\alpha_i \varphi_i}{\alpha_i + 1}, \quad \nu_{2_i} = \phi_i \left( \frac{\alpha_i \theta_i}{\alpha_i + q_i + 1} - 1 \right) - \frac{\varphi_i}{\Gamma(\alpha_i+2)},$$

$$\nu_{3_i} = \phi_i \left( 1 - \frac{\alpha_i \theta_i}{\alpha_i + q_i + 1} \right) - \frac{\varphi_i}{\Gamma(\alpha_i+2)},$$

$$\phi_i = \frac{\theta_i^* \theta_i^{\alpha_i+q_i}}{(1-\alpha_i)\Gamma(\alpha_i+q_i+1)}, \quad \varphi_i = \frac{\theta_i^* \theta_i^{q_i+1}}{(1-\alpha_i)\Gamma(q_i+2)},$$

$$\theta_i^* = \frac{\Gamma(q_i+1)}{\theta_i^{q_i} - \Gamma(q_i+1)}, \quad \theta_i^{q_i} \neq \Gamma(q_i+1).$$

**Proof.** By Lemma 2, we can write:

$$\begin{aligned}
y_i(t) &= \frac{1}{\Gamma(\alpha_i)} \int_0^t (t-s)^{\alpha_i-1} \left( \frac{1}{\Gamma(\beta_i)} \int_0^s (s-\tau)^{\beta_i-1} G_i(\tau) d\tau - \frac{k_i}{s^{\lambda_i}} x_i(s) \right) ds \\
&\quad - \frac{c_{0_i} t^{\alpha_i}}{\Gamma(\alpha_i+1)} - \frac{c_{1_i} t^{\alpha_i+1}}{\Gamma(\alpha_i+2)} - c_{2_i} t - c_{3_i}.
\end{aligned} \tag{4}$$

Hence, it yields that

$$\begin{aligned}
y_i'(t) &= \frac{1}{\Gamma(\alpha_i-1)} \int_0^t (t-s)^{\alpha_i-2} \left( \frac{1}{\Gamma(\beta_i)} \int_0^s (s-\tau)^{\beta_i-1} G_i(\tau) d\tau - \frac{k_i}{s^{\lambda_i}} x_i(s) \right) ds \\
&\quad - \frac{c_{0_i} t^{\alpha_i-1}}{\Gamma(\alpha_i)} - \frac{c_{1_i} t^{\alpha_i}}{\Gamma(\alpha_i+1)} - c_{2_i}.
\end{aligned} \tag{5}$$

And, then

$$y_i'(1) = y_i'(0) \quad \Rightarrow \quad \frac{c_{0_i}}{\Gamma(\alpha_i)} + \frac{c_{1_i}}{\Gamma(\alpha_i+1)} = I^{\alpha_i-1} R_i(1),$$

$$D^{\alpha_i} y_i(1) + \phi_{k_i,\lambda_i}(1) y_i(1) = \int_0^{\mu_i} T_i(\tau) y_i(\tau) d\tau \quad \Rightarrow \quad c_{0_i} + c_{1_i} = I^{\beta_i} G_i(1) - \int_0^{\mu_i} T_i(\tau) y_i(\tau) d\tau.$$

Therefore, we obtain

$$c_{0_i} = \frac{1}{1-\alpha_i} I^{\beta_i} G_i(1) - \frac{\Gamma(\alpha_i+1)}{1-\alpha_i} I^{\alpha_i-1} R_i(1) - \frac{1}{1-\alpha_i} \int_0^{\mu_i} T_i(\tau) y_i(\tau) d\tau,$$

$$c_{1_i} = -\frac{\alpha_i}{1-\alpha_i} I^{\beta_i} G_i(1) + \frac{\Gamma(\alpha_i+1)}{1-\alpha_i} I^{\alpha_i-1} R_i(1) + \frac{\alpha_i}{1-\alpha_i} \int_0^{\mu_i} T_i(\tau) y_i(\tau) d\tau.$$

Thanks to the boundary condition $y_i(1) = y_i(0)$, we conclude that

$$c_{2_i} = I^{\alpha_i} R_i(1) - \frac{1}{(1-\alpha_i)\Gamma(\alpha_i+1)} I^{\beta_i} G_i(1) + \frac{\alpha_i}{(1-\alpha)(1+\alpha)} I^{\alpha-1} R(1)$$
$$+ \frac{1}{(1-\alpha_i)\Gamma(\alpha_i+2)} \int_0^{\mu_i} T_i(\tau) y_i(\tau) d\tau.$$

By using the fact that $y_i(1) = y_i(0)$ and $I^{q_i} y_i(\theta_i) = y_i(1)$, we obtain

$$c_{3_i} = \theta_i^* \left[ I^{\alpha_i+q_i} R_i(\theta_i) - \frac{\theta_i^{\alpha_i+q_i}}{\Gamma(\alpha_i+q_i+1)} c_{0_i} - \frac{\theta_i^{\alpha_i+q_i+1}}{\Gamma(\alpha_i+q_i+2)} c_{1_i} - \frac{\theta_i^{q_i+1}}{\Gamma(q_i+2)} c_{2_i} \right]. \qquad (6)$$

Replacing $c_{0_i}, c_{1_i}, c_{2_i}$ in Equation (6), we get

$$c_{3_i} = \theta_i^* I^{\alpha_i+q_i} R_i(\theta_i) - \frac{\theta_i^* \theta_i^{q_i+1}}{\Gamma(q_i+2)} I^{\alpha_i} R_i(1) + \left[ \phi_i \left( \frac{\alpha_i \theta_i}{\alpha_i+q_i+1} - 1 \right) - \frac{\varphi_i}{\Gamma(\alpha_i+2)} \right] I^{\beta_i} G_i(1)$$
$$+ \left[ \phi_i \left( 1 - \frac{\theta_i \Gamma(\alpha_i+1)}{\alpha_i+q_i+1} \right) - \frac{\alpha_i \varphi_i}{\alpha_i+1} \right] I^{\alpha_i-1} R_i(1)) + \left[ \phi_i \left( 1 - \frac{\alpha_i \theta_i}{\alpha_i+q_i+1} \right) - \frac{\varphi_i}{\Gamma(\alpha_i+2)} \right]$$
$$\times \int_0^{\mu_i} T_i(\tau) y_i(\tau) d\tau.$$

Again, replacing $c_{0_i}, c_{1_i}, c_{2_i}, c_{3_i}$ in Equation (4), we obtain Equation (3).   □

We will use the fixed point theory to study the above singular system. To do this, we need to introduce the following notions.

We introduce the space:

$$X \times X := \{(x_1, x_2) \in C(J, \mathbb{R}) \times C(J, \mathbb{R}), D^{\delta_i} x_i \in C(J, \mathbb{R}), i = 1, 2\},$$

and the norm:

$$\|(x_1, x_2)\|_{X \times X} = Max\{\|x_1\|_\infty, \|x_2\|_\infty, \|D^{\delta_1} x_1\|_\infty, \|D^{\delta_2} x_2\|_\infty \},$$

where,

$$\|x_i\|_\infty = \sup_{t \in J} |x_i(t)|, \|D^{\delta_i} x_i\|_\infty = \sup_{t \in J} |D^{\delta_i} x_i(t)| \ ; i = 1, 2.$$

Then, we define the operator $S : X \times X \to X \times X$:

$$S(x_1, x_2) = (S_1(x_1, x_2), S_2(x_1, x_2))$$

such that, for any $t \in J$, we have

$$
\begin{aligned}
S_i(x_1, x_2)(t) = &\frac{1}{\Gamma(\alpha_i)} \int_0^t (t-s)^{\alpha_i-1} \left( \frac{1}{\Gamma(\beta_i)} \int_0^s (s-\tau)^{\beta_i-1} [l_i(\tau) - h_i(\tau, x_i(\tau)) - f_i(\tau, x_1(\tau), D^{\delta_i} x_2(\tau)) \right. \\
&\left. - g_i(\tau, x_1(\tau), I^{\rho_i} x_2(\tau))] d\tau - \frac{k_i}{s^{\lambda_i}} x_i(s) \right) ds + \left[ \frac{\theta_i^* \theta_i^{q_i+1}}{\Gamma(q_i+2)} - t \right] \frac{1}{\Gamma(\alpha_i)} \int_0^1 (1-s)^{\alpha_i-1} \\
&\times \left( \frac{1}{\Gamma(\beta_i)} \int_0^s (s-\tau)^{\beta_i-1} [l_i(\tau) - h_i(\tau, x_i(\tau)) - f_i(\tau, x_1(\tau), D^{\delta_i} x_2(\tau)) \right. \\
&\left. - g_i(\tau, x_1(\tau), I^{\rho_i} x_2(\tau))] d\tau - \frac{k_i}{s^{\lambda_i}} x_i(s) \right) ds + \left[ \frac{t^{\alpha_i}}{1-\alpha_i} - \frac{t^{\alpha_i+1} + \alpha_i t}{(1-\alpha_i)(1+\alpha_i)} \right. \\
&\left. - \nu_{1_i} \right] \frac{1}{\Gamma(\alpha_i-1)} \int_0^1 (1-s)^{\alpha_i-2} \left( \frac{1}{\Gamma(\beta_i)} \int_0^s (s-\tau)^{\beta_i-1} \right. \\
&\left. \times [l_i(\tau) - h_i(\tau, x_i(\tau)) - f_i(\tau, x_1(\tau), D^{\delta_i} x_2(\tau)) - g_i(\tau, x_1(\tau), I^{\rho_i} x_2(\tau))] d\tau - \frac{k_i}{s^{\lambda_i}} x_i(s) \right) ds \\
&- \frac{1}{\Gamma(\alpha_i+q_i)} \int_0^{\theta_i} (\theta_i - s)^{\alpha_i+q_i-1} \left( \frac{1}{\Gamma(\beta_i)} \int_0^s (s-\tau)^{\beta_i-1} [l_i(\tau) - h_i(\tau, x_i(\tau)) \right. \\
&\left. - f_i(\tau, x_1(\tau), D^{\delta_i} x_2(\tau)) - g_i(\tau, x_1(\tau), I^{\rho_i} x_2(\tau))] d\tau - \frac{k_i}{s^{\lambda_i}} x_i(s) \right) ds \\
&+ \left[ \frac{t - t^{\alpha_i+1}}{(1-\alpha_i)\Gamma(\alpha_i+1)} + \frac{\alpha_i t^{\alpha_i+1}}{(1-\alpha_i)\Gamma(\alpha_i+2)} - \nu_{2_i} \right] \\
&\times \frac{1}{\Gamma(\beta_i)} \int_0^1 (1-\tau)^{\beta_i-1} [l_i(\tau) - h_i(\tau, x_i(\tau)) \\
&- f_i(\tau, x_1(\tau), D^{\delta_i} x_2(\tau)) - g_i(\tau, x_1(\tau), I^{\rho_i} x_2(\tau))] d\tau \\
&+ \left[ \frac{t^{\alpha_i} - \alpha_i t^{\alpha_i+1}}{(1-\alpha_i)\Gamma(\alpha_i+1)} - \frac{t}{(1-\alpha_i)\Gamma(\alpha_i+2)} - \nu_{3_i} \right] \int_0^{\mu_i} T_i(\tau) x_i(\tau) d\tau.
\end{aligned}
\tag{7}
$$

At the end of this section, we are ready to present our main results.

## 3. Main Results

First of all, we shall take into account the following hypotheses:

$(T1)$ : The functions $f_i$ and $g_i$ are continuous over $J \times \mathbb{R}^2$,
   $h_i$ are continuous over $J \times \mathbb{R}$ and $l_i$ are continuous over $J$; $i = 1, 2$.

$(T2)$ : There are nonnegative constants $w_{ij}$ and $w'_{ij}$; $i = 1, 2, j = 1, 2$, such that:
   for any $t \in J$, $x_i, y_i \in \mathbb{R}$

$$
\begin{aligned}
|f_i(t, x_1, x_2) - f_i(t, y_1, y_2)| &\leq \sum_{j=1}^2 w_{ij} |x_j - y_j| \ ; \ i = 1, 2, \\
|g_i(t, x_1, x_2) - g_i(t, y_1, y_2)| &\leq \sum_{j=1}^2 w'_{ij} |x_j - y_j| \ ; \ i = 1, 2.
\end{aligned}
\tag{8}
$$

and for any $t \in J$, $x, y \in \mathbb{R}$

$$
|h_i(t, x) - h_i(t, y)| \leq r_i |x - y| \ ; \ i = 1, 2.
\tag{9}
$$

We pose

$$\Delta_{f_i} = \max_{j=1,2} \left\{ w_{ij} \right\},$$

$$\Delta_{g_i} = \max_{j=1,2} \left\{ w'_{ij} \right\}.$$

$(T3)$ : There exist non negative constants $N_{1_i}, N_{2_i}, N_{3_i}$, that satisfy for any $t \in J, x \in \mathbb{R}^2, y \in \mathbb{R}$

$$|f_i(t,x)| \le N_{1_i}, \quad |g_i(t,x)| \le N_{2_i}, \quad |h_i(t,y)| \quad \le N_{3_i}.$$

$(T4)$ : We take $\|l_i\|_\infty = N_{4_i}; i = 1, 2$. Then, we consider the quantities:

$$
\begin{aligned}
\varepsilon_i 1 &= \left[ r_i + 2\Delta_{f_i} + \Delta_{g_i} + \frac{\Delta_{g_i}}{\Gamma(\rho_i + 1)} \right] \left[ \left( 2 + \frac{|\theta_i^*| \theta_i^{q_i+1}}{\Gamma(q_i + 2)} \right) \frac{1}{\Gamma(\alpha_i + \beta_i + 1)} \right. \\
&+ \left( \frac{2}{(\alpha_i - 1)} + |\nu_{1_i}| \right) \frac{1}{\Gamma(\alpha_i + \beta_i)} + \frac{\theta^{\alpha_i + \beta_i + q_i}}{\Gamma(\alpha_i + \beta_i + q_i + 1)} \\
&+ \left( \frac{2}{(\alpha_i - 1)\Gamma(\alpha_i + 1)} + \frac{\alpha_i}{(\alpha_i - 1)\Gamma(\alpha_i + 2)} + |\nu_{2_i}| \right) \frac{1}{\Gamma(\beta_i + 1)} \right] \\[1em]
&+ \left( \frac{1 + \alpha_i}{(\alpha_i - 1)\Gamma(\alpha_i + 1)} + \frac{1}{(\alpha_i - 1)\Gamma(\alpha_i + 2)} + |\nu_{3_i}| \right) \chi_i \mu_i + k_i \Gamma(1 - \lambda_i) \\
&\times \left[ \left( 2 + \frac{|\theta_i^*| \theta_i^{q_i+1}}{\Gamma(q_i + 2)} \right) \frac{1}{\Gamma(\alpha_i - \lambda_i + 1)} + \left( \frac{2}{(\alpha_i - 1)} + |\nu_{1_i}| \right) \frac{1}{\Gamma(\alpha_i - \lambda_i)} \right. \\
&+ \left. \frac{1}{\Gamma(\alpha_i - \lambda_i + q_i + 1)} \right],
\end{aligned}
$$

and

$$
\begin{aligned}
\varepsilon_i 2 &= \left[ r_i + 2\Delta_{f_i} + \Delta_{g_i} + \frac{\Delta_{g_i}}{\Gamma(\rho_i + 1)} \right] \left[ \frac{1}{\Gamma(\alpha_i - \delta_i + \beta_i + 1)} + \frac{1}{\Gamma(\alpha_i + \beta_i + 1)\Gamma(2 - \delta_i)} \right. \\
&+ \left( \frac{\Gamma(\alpha_i + 1)}{(\alpha_i - 1)\Gamma(\alpha_i - \delta_i + 1)} + \frac{\Gamma(\alpha_i + 1)}{(\alpha_i - 1)\Gamma(\alpha_i - \delta_i + 2)} \right. \\
&+ \left. \frac{\alpha_i}{(\alpha_i - 1)(1 + \alpha_i)\Gamma(2 - \delta_i)} \right) \frac{1}{\Gamma(\alpha_i + \beta_i)} \\
&+ \left. \left( \frac{1}{(\alpha_i - 1)\Gamma(\alpha_i + 1)\Gamma(2 - \delta_i)} + \frac{1}{(\alpha_i - 1)\Gamma(\alpha_i - \delta_i + 2)} \right) \frac{1}{\Gamma(\beta_i + 1)} \right] \\
&+ \left( \frac{1}{(\alpha_i - 1)\Gamma(\alpha_i - \delta_i + 1)} + \frac{\alpha_i(\alpha_i + 1)}{(\alpha_i - 1)\Gamma(\alpha_i - \delta_i + 2)} + \frac{1}{(\alpha_i - 1)\Gamma(\alpha_i + 2)\Gamma(2 - \delta_i)} \right) \chi_i \mu_i \\[1em]
&+ k_i \Gamma(1 - \lambda_i) \left[ \frac{1}{\Gamma(\alpha_i - \delta_i - \lambda_i + 1)} + \frac{1}{\Gamma(\alpha_i - \lambda_i + 1)\Gamma(2 - \delta_i)} \right. \\
&+ \left( \frac{\Gamma(\alpha_i + 1)}{(\alpha_i - 1)\Gamma(\alpha_i - \delta_i + 1)} + \frac{\Gamma(\alpha_i + 1)}{(\alpha_i - 1)\Gamma(\alpha_i - \delta_i + 2)} \right. \\
&+ \left. \left. \frac{\alpha_i}{(\alpha_i - 1)(1 + \alpha_i)\Gamma(2 - \delta_i)} \right) \frac{1}{\Gamma(\alpha_i - \lambda_i)} \right],
\end{aligned}
$$

The first main result deals with the existence of a unique solution for Equation (1). We prove the following first main result:

**Theorem 1.** *Assume that $(T i)_{i=2,3}$ are satisfied. Then, the problem (1) has a unique solution, provided that $\varepsilon_i < 1$, where $\varepsilon_i = \max \{\varepsilon_i 1, \varepsilon_i 2\}$.*

**Proof.** We proceed to prove that $S$ is a contraction mapping. For $(x, y) \in X \times X$, we can write

$$\|S_i(y_1, y_2) - S_i(x_1, x_2)\|_\infty \leq \left[ r_i + 2\Delta_{f_i} + \Delta_{g_i} + \frac{\Delta_{g_i}}{\Gamma(\rho_i + 1)} \right] \left[ \left( 2 + \frac{|\theta_i^*|\theta_i^{q_i+1}}{\Gamma(q_i + 2)} \right) \frac{1}{\Gamma(\alpha_i + \beta_i + 1)} \right.$$

$$+ \left( \frac{2}{(\alpha_i - 1)} + |\nu_{1_i}| \right) \frac{1}{\Gamma(\alpha_i + \beta_i)} + \frac{\theta^{\alpha_i + \beta_i + q_i}}{\Gamma(\alpha_i + \beta_i + q_i + 1)}$$

$$+ \left( \frac{2}{(\alpha_i - 1)\Gamma(\alpha_i + 1)} + \frac{\alpha_i}{(\alpha_i - 1)\Gamma(\alpha_i + 2)} + |\nu_{2_i}| \right) \frac{1}{\Gamma(\beta_i + 1)} \right]$$

$$\times \|(y_1, y_2) - (x_1, x_2)\|_{X \times X}$$

$$+ \left( \frac{1 + \alpha_i}{(\alpha_i - 1)\Gamma(\alpha_i + 1)} + \frac{1}{(\alpha_i - 1)\Gamma(\alpha_i + 2)} + |\nu_{3_i}| \right) \chi_i \mu_i$$

$$\times \|(y_1, y_2) - (x_1, x_2)\|_{X \times X}$$

$$+ k_i \Gamma(1 - \lambda_i) \left[ \left( 2 + \frac{|\theta_i^*|\theta_i^{q_i+1}}{\Gamma(q_i + 2)} \right) \frac{1}{\Gamma(\alpha_i - \lambda_i + 1)} \right.$$

$$+ \left( \frac{2}{(\alpha_i - 1)} + |\nu_{1_i}| \right) \frac{1}{\Gamma(\alpha_i - \lambda_i)}$$

$$+ \frac{1}{\Gamma(\alpha_i - \lambda_i + q_i + 1)} \right] \|(y_1, y_2) - (x_1, x_2)\|_{X \times X}.$$

We have:

$$D^{\delta_i} S_i(y_1, y_2)(t)$$
$$= \frac{1}{\Gamma(\alpha_i - \delta_i)} \int_0^t (t - s)^{\alpha_i - \delta_i - 1} \left( \frac{1}{\Gamma(\beta_i)} \int_0^s (s - \tau)^{\beta_i - 1} [l_i(\tau) - h_i(\tau, x_i(\tau)) - f_i(\tau, x_1(\tau), D^{\delta_i} x_2(\tau)) \right.$$

$$\left. - g_i(\tau, x_1(\tau), I^{\rho_i} x_2(\tau))] d\tau - \frac{k_i}{s^{\lambda_i}} x_i(s) \right) ds - \frac{t^{1 - \delta_i}}{\Gamma(2 - \delta_i)\Gamma(\alpha_i)} \int_0^1 (1 - s)^{\alpha_i - 1} \left( \frac{1}{\Gamma(\beta_i)} \int_0^s (s - \tau)^{\beta_i - 1} \right.$$

$$\left. \times [l_i(\tau) - h_i(\tau, x_i(\tau)) - f_i(\tau, x_1(\tau), D^{\delta_i} x_2(\tau)) - g_i(\tau, x_1(\tau), I^{\rho_i} x_2(\tau))] d\tau - \frac{k_i}{s^{\lambda_i}} x_i(s) \right) ds$$

$$+ \left[ \frac{\Gamma(\alpha_i + 1) t^{\alpha_i - \delta_i}}{(1 - \alpha_i)\Gamma(\alpha_i - \delta_i + 1)} - \frac{\Gamma(\alpha_i + 1) t^{\alpha_i - \delta_i + 1}}{(1 - \alpha_i)\Gamma(\alpha_i - \delta_i + 2)} - \frac{\alpha_i t^{1 - \delta_i}}{(1 - \alpha_i)(1 + \alpha_i)\Gamma(2 - \delta_i)} \right] \frac{1}{\Gamma(\alpha_i - 1)} \quad (10)$$

$$\times \int_0^1 (1 - s)^{\alpha_i - 2} \left( \frac{1}{\Gamma(\beta_i)} \int_0^s (s - \tau)^{\beta_i - 1} [l_i(\tau) - h_i(\tau, x_i(\tau)) - f_i(\tau, x_1(\tau), D^{\delta_i} x_2(\tau)) - g_i(\tau, x_1(\tau), \right.$$

$$\left. I^{\rho_i} x_2(\tau))] d\tau - \frac{k_i}{s^{\lambda_i}} x_i(s) \right) ds + \left[ \frac{t^{1 - \delta_i}}{(1 - \alpha_i)\Gamma(1 + \alpha_i)\Gamma(2 - \delta_i)} - \frac{t^{\alpha_i - \delta_i + 1}}{(1 - \alpha_i)\Gamma(\alpha_i - \delta_i + 2)} \right] \frac{1}{\Gamma(\beta_i)}$$

$$\times \int_0^1 (1 - \tau)^{\beta_i - 1} [l_i(\tau) - h_i(\tau, x_i(\tau)) - f_i(\tau, x_1(\tau), D^{\delta_i} x_2(\tau)) - g_i(\tau, x_1(\tau), I^{\rho_i} x_2(\tau))] d\tau$$

$$+ \left[ \frac{t^{\alpha_i - \delta_i}}{(1 - \alpha_i)\Gamma(\alpha_i - \delta_i + 1)} - \frac{\alpha_i(\alpha_i + 1) t^{\alpha_i - \delta_i + 1}}{(1 - \alpha_i)\Gamma(\alpha_i - \delta_i + 2)} - \frac{t^{1 - \delta_i}}{(1 - \alpha_i)\Gamma(\alpha_i + 2)\Gamma(2 - \delta_i)} \right] \int_0^{\mu_i} T_i(\tau) x_i(\tau) d\tau.$$

On the other hand, we can write

$$\|D^{\delta_i}S_i(y_1, y_2) - D^{\delta_i}S_i(x_1, x_2)\|_\infty$$

$$\leq \left[ r_i + 2\Delta_{f_i} + \Delta_{g_i} + \frac{\Delta_{g_i}}{\Gamma(\rho_i+1)} \right] \left[ \frac{1}{\Gamma(\alpha_i - \delta_i + \beta_i + 1)} + \frac{1}{\Gamma(\alpha_i + \beta_i + 1)\Gamma(2 - \delta_i)} \right.$$

$$+ \left( \frac{\Gamma(\alpha_i + 1)}{(\alpha_i - 1)\Gamma(\alpha_i - \delta_i + 1)} + \frac{\Gamma(\alpha_i + 1)}{(\alpha_i - 1)\Gamma(\alpha_i - \delta_i + 2)} + \frac{\alpha_i}{(\alpha_i - 1)(1 + \alpha_i)\Gamma(2 - \delta_i)} \right) \frac{1}{\Gamma(\alpha_i + \beta_i)}$$

$$\left. + \left( \frac{1}{(\alpha_i - 1)\Gamma(\alpha_i + 1)\Gamma(2 - \delta_i)} + \frac{1}{(\alpha_i - 1)\Gamma(\alpha_i - \delta_i + 2)} \right) \frac{1}{\Gamma(\beta_i + 1)} \right] \|(y_1, y_2) - (x_1, x_2)\|_{X \times X}$$

$$+ \left( \frac{1}{(\alpha_i - 1)\Gamma(\alpha_i - \delta_i + 1)} + \frac{\alpha_i(\alpha_i + 1)}{(\alpha_i - 1)\Gamma(\alpha_i - \delta_i + 2)} + \frac{1}{(\alpha_i - 1)\Gamma(\alpha_i + 2)\Gamma(2 - \delta_i)} \right) \chi_i \mu_i$$

$$\times \|(y_1, y_2) - (x_1, x_2)\|_{X \times X}$$

$$+ k_i \Gamma(1 - \lambda_i) \left[ \frac{1}{\Gamma(\alpha_i - \delta_i - \lambda_i + 1)} + \frac{1}{\Gamma(\alpha_i - \lambda_i + 1)\Gamma(2 - \delta_i)} \right.$$

$$\left. + \left( \frac{\Gamma(\alpha_i + 1)}{(\alpha_i - 1)\Gamma(\alpha_i - \delta_i + 1)} + \frac{\Gamma(\alpha_i + 1)}{(\alpha_i - 1)\Gamma(\alpha_i - \delta_i + 2)} + \frac{\alpha_i}{(\alpha_i - 1)(1 + \alpha_i)\Gamma(2 - \delta_i)} \right) \frac{1}{\Gamma(\alpha_i - \lambda_i)} \right]$$

$$\times \|(y_1, y_2) - (x_1, x_2)\|_{X \times X}.$$

Consequently,

$$\|S_i(y_1, y_2) - S_i(x_1, x_2)\|_{X \times X} \leq \varepsilon_i \|(y_1, y_2) - (x_1, x_2)\|_{X \times X}.$$

Using the condition on $\varepsilon_i$; $i = 1, 2$, we conclude that $S$ is contractive. So applying Banach contraction principle, we see that $S$ admits a unique fixed point $(x_1, x_2)$ which is the solution of the problem (1).　□

The following main result deals with the existence of at least one solution.

**Theorem 2.** *Assume that hypotheses* $(T1)$, $(T3)$ *and* $(T4)$ *are satisfied. Then,* (1) *has at least one solution defined over J.*

**Proof.** Let us prove the result by applying Schaefer fixed point theorem. We do this by considering the following steps:

**Step 1:** Since the functions $f_i, g_i, h_i$ and $l_i, i = 1, 2$ are continuous, then $S$ is continuous on $X \times X$.

**Step 2:** $S$ maps bounded sets into bounded sets in $X \times X$:

Let us take $r > 0$ and $B_r := \{(x_1, x_2) \in X \times X; \|(x_1, x_2)\|_{X \times X} \leq r\}$. For $(y_1, y_2) \in B_r$, thanks to $(T3)$ and $(T4)$, we can write

$$\|S_i(y_1, y_2)\|_\infty$$

$$
\leq \left[N_{1_i} + N_{2_i} + N_{3_i} + N_{4_i}\right]\left[\left(2 + \frac{|\theta_i^*||\theta_i^{q_i+1}}{\Gamma(q_i+2)}\right)\frac{1}{\Gamma(\alpha_i+\beta_i+1)}\right.
$$

$$
+ \left(\frac{2}{(\alpha_i-1)} + |\nu_{1_i}|\right)\frac{1}{\Gamma(\alpha_i+\beta_i)} + \frac{\theta^{\alpha_i+\beta_i+q_i}}{\Gamma(\alpha_i+\beta_i+q_i+1)}
$$

$$
\left.+ \left(\frac{2}{(\alpha_i-1)\Gamma(\alpha_i+1)} + \frac{\alpha_i}{(\alpha_i-1)\Gamma(\alpha_i+2)} + |\nu_{2_i}|\right)\frac{1}{\Gamma(\beta_i+1)}\right]
$$

$$
+ \left(\frac{1+\alpha_i}{(\alpha_i-1)\Gamma(\alpha_i+1)} + \frac{1}{(\alpha_i-1)\Gamma(\alpha_i+2)} + |\nu_{3_i}|\right)r\chi_i\mu_i \tag{11}
$$

$$
+ k_i r\Gamma(1-\lambda_i)\left[\left(2 + \frac{|\theta_i^*||\theta_i^{q_i+1}}{\Gamma(q_i+2)}\right)\frac{1}{\Gamma(\alpha_i-\lambda_i+1)} + \left(\frac{2}{(\alpha_i-1)} + |\nu_{1_i}|\right)\frac{1}{\Gamma(\alpha_i-\lambda_i)}\right.
$$

$$
\left.+ \frac{1}{\Gamma(\alpha_i-\lambda_i+q_i+1)}\right] < +\infty
$$

and

$$\|D^{\delta_i}S_i(y_1, y_2)\|_\infty$$

$$
\leq \left[N_{1_i} + N_{2_i} + N_{3_i} + N_{4_i}\right]\left[\frac{1}{\Gamma(\alpha_i-\delta_i+\beta_i+1)} + \frac{1}{\Gamma(\alpha_i+\beta_i+1)\Gamma(2-\delta_i)}\right.
$$

$$
+ \left(\frac{\Gamma(\alpha_i+1)}{(\alpha_i-1)\Gamma(\alpha_i-\delta_i+1)} + \frac{\Gamma(\alpha_i+1)}{(\alpha_i-1)\Gamma(\alpha_i-\delta_i+2)} + \frac{\alpha_i}{(\alpha_i-1)(1+\alpha_i)\Gamma(2-\delta_i)}\right)\frac{1}{\Gamma(\alpha_i+\beta_i)}
$$

$$
\left.+ \left(\frac{1}{(\alpha_i-1)\Gamma(\alpha_i+1)\Gamma(2-\delta_i)} + \frac{1}{(\alpha_i-1)\Gamma(\alpha_i-\delta_i+2)}\right)\frac{1}{\Gamma(\beta_i+1)}\right] \tag{12}
$$

$$
+ \left(\frac{1}{(\alpha_i-1)\Gamma(\alpha_i-\delta_i+1)} + \frac{\alpha_i(\alpha_i+1)}{(\alpha_i-1)\Gamma(\alpha_i-\delta_i+2)} + \frac{1}{(\alpha_i-1)\Gamma(\alpha_i+2)\Gamma(2-\delta_i)}\right)r\chi_i\mu_i
$$

$$
+ k_i r\Gamma(1-\lambda_i)\left[\frac{1}{\Gamma(\alpha_i-\delta_i-\lambda_i+1)} + \frac{1}{\Gamma(\alpha_i-\lambda_i+1)\Gamma(2-\delta_i)} + \left(\frac{\Gamma(\alpha_i+1)}{(\alpha_i-1)\Gamma(\alpha_i-\delta_i+1)}\right.\right.
$$

$$
\left.\left.+ \frac{\Gamma(\alpha_i+1)}{(\alpha_i-1)\Gamma(\alpha_i-\delta_i+2)} + \frac{\alpha_i}{(\alpha_i-1)(1+\alpha_i)\Gamma(2-\delta_i)}\right)\frac{1}{\Gamma(\alpha_i-\lambda_i)}\right] < +\infty.
$$

So, for any $(y_1, y_2) \in B_r$, we have $\|S(y_1, y_2)\|_{X \times X} < +\infty$.

Consequently, $S$ is uniformly bounded on $B_r$.

**Step 3:** $S$ maps bounded sets into equicontinuous sets of $X \times X$ :

Let $t_1, t_2 \in J, t_1 < t_2$ and let $B_r$ be the above bounded set of $X \times X$. So by considering $x = (x_1, x_2) \in B_r$, we can state that for each $t \in J$, we have

$$|S_i x(t_2) - S_i x(t_1)|$$

$$\leq \quad |\frac{1}{\Gamma(\alpha_i)} \int_0^{t_2} (t_2 - s)^{\alpha_i - 1} \left( \frac{1}{\Gamma(\beta_i)} \int_0^s (s - \tau)^{\beta_i - 1} [l_i(\tau) - h_i(\tau, x_i(\tau)) - f_i(\tau, x_1(\tau), D^{\delta_i} x_2(\tau))$$

$$- g_i(\tau, x_1(\tau), I^{\rho_i} x_2(\tau))] d\tau - \frac{k_i}{s^{\lambda_i}} x_i(s) \right) ds - \frac{1}{\Gamma(\alpha_i)} \int_0^{t_1} (t_1 - s)^{\alpha_i - 1} \left( \frac{1}{\Gamma(\beta_i)} \int_0^s (s - \tau)^{\beta_i - 1} [l_i(\tau)$$

$$- h_i(\tau, x_i(\tau)) - f_i(\tau, x_1(\tau), D^{\delta_i} x_2(\tau)) - g_i(\tau, x_1(\tau), I^{\rho_i} x_2(\tau))] d\tau - \frac{k_i}{s^{\lambda_i}} x_i(s) \right) ds|$$

$$+ \left[ \frac{N_{1_i} + N_{2_i} + N_{3_i} + N_{4_i}}{\Gamma(\alpha_i + \beta_i + 1)} + + k_i r \frac{\Gamma(1 - \lambda_i)}{\Gamma(1 - \lambda_i + \alpha_i)} \right] |t_1 - t_2|$$

$$+ \left[ \frac{N_{1_i} + N_{2_i} + N_{3_i} + N_{4_i}}{\Gamma(\alpha_i + \beta_i)} + k_i r \frac{\Gamma(1 - \lambda_i)}{\Gamma(\alpha_i - \lambda_i)} \right] \left[ \frac{|t_1^{\alpha_i} - t_2^{\alpha_i}|}{\alpha_i - 1} + \frac{|t_1^{\alpha_i + 1} - t_2^{\alpha_i + 1}| + \alpha_i |t_1 - t_2|}{(\alpha_i - 1)(\alpha_i + 1)} \right]$$

$$+ \frac{N_{1_i} + N_{2_i} + N_{3_i} + N_{4_i}}{\Gamma(\beta + 1)} \left[ \frac{|t_1 - t_2| + |t_1^{\alpha_i + 1} - t_2^{\alpha_i + 1}|}{(\alpha_i - 1)\Gamma(\alpha_i + 1)} + \frac{\alpha_i |t_1^{\alpha_i + 1} - t_2^{\alpha_i + 1}|}{(\alpha_i - 1)\Gamma(\alpha_i + 2)} \right]$$

$$+ \left[ \frac{|t_1^{\alpha_i} - t_2^{\alpha_i}| + \alpha_i |t_1^{\alpha_i + 1} - t_2^{\alpha_i + 1}|}{(\alpha_i - 1)\Gamma(\alpha_i + 1)} + \frac{|t_1 - t_2|}{(\alpha_i - 1)\Gamma(\alpha_i + 2)} \right] r \chi_i \mu_i,$$

$$\leq \quad |\frac{1}{\Gamma(\alpha_i)} \int_0^{t_1} (t_2 - s)^{\alpha_i - 1} \left( \frac{1}{\Gamma(\beta_i)} \int_0^s (s - \tau)^{\beta_i - 1} [l_i(\tau) - h_i(\tau, x_i(\tau)) - f_i(\tau, x_1(\tau), D^{\delta_i} x_2(\tau))$$

$$- g_i(\tau, x_1(\tau), I^{\rho_i} x_2(\tau))] d\tau - \frac{k_i}{s^{\lambda_i}} x_i(s) \right) ds - \frac{1}{\Gamma(\alpha_i)} \int_0^{t_1} (t_1 - s)^{\alpha_i - 1} \left( \frac{1}{\Gamma(\beta_i)} \int_0^s (s - \tau)^{\beta_i - 1} [l_i(\tau)$$

$$- h_i(\tau, x_i(\tau)) - f_i(\tau, x_1(\tau), D^{\delta_i} x_2(\tau)) - g_i(\tau, x_1(\tau), I^{\rho_i} x_2(\tau))] d\tau - \frac{k_i}{s^{\lambda_i}} x_i(s) \right) ds|$$

$$+ |\frac{1}{\Gamma(\alpha_i)} \int_{t_1}^{t_2} (t_2 - s)^{\alpha_i - 1} \left( \frac{1}{\Gamma(\beta_i)} \int_0^s (s - \tau)^{\beta_i - 1} [l_i(\tau) - h_i(\tau, x_i(\tau)) - f_i(\tau, x_1(\tau), D^{\delta_i} x_2(\tau))$$

$$\quad (13)$$

$$- g_i(\tau, x_1(\tau), I^{\rho_i} x_2(\tau))] d\tau - \frac{k_i}{s^{\lambda_i}} x_i(s) \right) ds|$$

$$+ \left[ \frac{N_{1_i} + N_{2_i} + N_{3_i} + N_{4_i}}{\Gamma(\alpha_i + \beta_i + 1)} + + k_i r \frac{\Gamma(1 - \lambda_i)}{\Gamma(1 - \lambda_i + \alpha_i)} \right] |t_1 - t_2|$$

$$+ \left[ \frac{N_{1_i} + N_{2_i} + N_{3_i} + N_{4_i}}{\Gamma(\alpha_i + \beta_i)} + k_i r \frac{\Gamma(1 - \lambda_i)}{\Gamma(\alpha_i - \lambda_i)} \right] \left[ \frac{|t_1^{\alpha_i} - t_2^{\alpha_i}|}{\alpha_i - 1} + \frac{|t_1^{\alpha_i + 1} - t_2^{\alpha_i + 1}| + \alpha_i |t_1 - t_2|}{(\alpha_i - 1)(\alpha_i + 1)} \right]$$

$$+ \frac{N_{1_i} + N_{2_i} + N_{3_i} + N_{4_i}}{\Gamma(\beta + 1)} \left[ \frac{|t_1 - t_2| + |t_1^{\alpha_i + 1} - t_2^{\alpha_i + 1}|}{(\alpha_i - 1)\Gamma(\alpha_i + 1)} + \frac{\alpha_i |t_1^{\alpha_i + 1} - t_2^{\alpha_i + 1}|}{(\alpha_i - 1)\Gamma(\alpha_i + 2)} \right]$$

$$+ \left[ \frac{|t_1^{\alpha_i} - t_2^{\alpha_i}| + \alpha_i |t_1^{\alpha_i + 1} - t_2^{\alpha_i + 1}|}{(\alpha_i - 1)\Gamma(\alpha_i + 1)} + \frac{|t_1 - t_2|}{(\alpha_i - 1)\Gamma(\alpha_i + 2)} \right] r \chi_i \mu_i,$$

$$\leq \quad \frac{N_{1_i} + N_{2_i} + N_{3_i} + N_{4_i}}{\Gamma(\alpha_i + \beta_i + 1)} \left[ |t_1^{\beta_i + \alpha_i} - t_2^{\beta_i + \alpha_i}| + 2|t_1 - t_2|^{\beta_i + \alpha_i} + |t_1 - t_2| \right]$$

$$+ k_i r \frac{\Gamma(1 - \lambda_i)}{\Gamma(1 - \lambda_i + \alpha_i)} \left[ |t_1^{\alpha_i - \lambda_i} - t_2^{\alpha_i - \lambda_i}| + |t_1 - t_2| \right]$$

$$+ \left[ \frac{N_{1_i} + N_{2_i} + N_{3_i} + N_{4_i}}{\Gamma(\alpha_i + \beta_i)} + k_i r \frac{\Gamma(1 - \lambda_i)}{\Gamma(\alpha_i - \lambda_i)} \right] \left[ \frac{|t_1^{\alpha_i} - t_2^{\alpha_i}|}{\alpha_i - 1} + \frac{|t_1^{\alpha_i + 1} - t_2^{\alpha_i + 1}| + \alpha_i |t_1 - t_2|}{(\alpha_i - 1)(\alpha_i + 1)} \right]$$

$$+ \frac{N_{1_i} + N_{2_i} + N_{3_i} + N_{4_i}}{\Gamma(\beta + 1)} \left[ \frac{|t_1 - t_2| + |t_1^{\alpha_i + 1} - t_2^{\alpha_i + 1}|}{(\alpha_i - 1)\Gamma(\alpha_i + 1)} + \frac{\alpha_i |t_1^{\alpha_i + 1} - t_2^{\alpha_i + 1}|}{(\alpha_i - 1)\Gamma(\alpha_i + 2)} \right]$$

$$+ \left[ \frac{|t_1^{\alpha_i} - t_2^{\alpha_i}| + \alpha_i |t_1^{\alpha_i + 1} - t_2^{\alpha_i + 1}|}{(\alpha_i - 1)\Gamma(\alpha_i + 1)} + \frac{|t_1 - t_2|}{(\alpha_i - 1)\Gamma(\alpha_i + 2)} \right] r \chi_i \mu_i.$$

With the same arguments as before, we have

$$|D^{\delta_i} S_i x(t_2) - D^{\delta_i} S_i x(t_1)|$$

$$
\begin{aligned}
\leq \quad & \frac{N_{1_i} + N_{2_i} + N_{3_i} + N_{4_i}}{\Gamma(\alpha_i - \delta_i + \beta_i + 1)} \left[ |t_1^{\beta_i + \alpha_i - \delta_i} - t_2^{\beta_i + \alpha_i - \delta_i}| + 2|t_1 - t_2|^{\beta_i + \alpha_i - \delta_i} \right] \\
& + k_i r \frac{\Gamma(1 - \lambda_i)}{\Gamma(1 - \lambda_i + \alpha_i - \delta_i)} |t_1^{\alpha_i - \lambda_i - \delta_i} - t_2^{\alpha_i - \lambda_i - \delta_i}| \\
& + \left[ \frac{N_{1_i} + N_{2_i} + N_{3_i} + N_{4_i}}{\Gamma(\alpha_i + \beta_i + 1)} + k_i r \frac{\Gamma(1 - \lambda_i)}{\Gamma(\alpha_i - \lambda_i + 1)} \right] \frac{|t_1^{1 - \delta_i} - t_2^{1 - \delta_i}|}{\Gamma(2 - \delta_i)} \\[1em]
& + \left[ \frac{N_{1_i} + N_{2_i} + N_{3_i} + N_{4_i}}{\Gamma(\alpha_i + \beta_i)} + k_i r \frac{\Gamma(1 - \lambda_i)}{\Gamma(\alpha_i - \lambda_i)} \right] \left[ \frac{\Gamma(\alpha_i + 1)|t_1^{\alpha_i - \delta_i} - t_2^{\alpha_i - \delta_i}|}{(\alpha_i - 1)\Gamma(\alpha_i - \delta_i + 1)} \right. \\
& \left. + \frac{\Gamma(\alpha_i + 1)|t_1^{\alpha_i - \delta_i + 1} - t_2^{\alpha_i - \delta_i + 1}|}{(\alpha_i - 1)\Gamma(\alpha_i - \delta_i + 2)} + \frac{\alpha_i |t_1^{1 - \delta_i} - t_2^{1 - \delta_i}|}{(\alpha_i - 1)(\alpha_i + 1)\Gamma(2 - \delta_i)} \right] \\
& + \frac{N_{1_i} + N_{2_i} + N_{3_i} + N_{4_i}}{\Gamma(\beta_i + 1)} \left[ \frac{|t_1^{1 - \delta_i} - t_2^{1 - \delta_i}|}{(\alpha_i - 1)\Gamma(\alpha_i + 1)\Gamma(2 - \delta_i)} + \frac{|t_1^{\alpha_i - \delta_i + 1} - t_2^{\alpha_i - \delta_i + 1}|}{(\alpha_i - 1)\Gamma(\alpha_i - \delta_i + 2)} \right] \\[1em]
& + \left[ \frac{|t_1^{\alpha_i - \delta_i} - t_2^{\alpha_i - \delta_i}|}{(\alpha_i - 1)\Gamma(\alpha_i - \delta_i + 1)} + \frac{\alpha_i(\alpha_i + 1)|t_1^{\alpha_i - \delta_i + 1} - t_2^{\alpha_i - \delta_i + 1}|}{(\alpha_i - 1)\Gamma(\alpha_i - \delta_i + 2)} \right. \\
& \left. + \frac{|t_1^{1 - \delta_i} - t_2^{1 - \delta_i}|}{(\alpha_i - 1)\Gamma(\alpha_i + 2)\Gamma(2 - \delta_i)} \right] r \chi_i \mu_i.
\end{aligned}
\tag{14}
$$

The right hand sides of Equations (13) and (14) tend to zero independently of $x$ as $t_1 \to t_2$.

As a consequence of Steps 1–3 and thanks to Ascoli-Arzela theorem, we conclude that $S$ is completely continuous.

**Step 4:** The set $A := \{(x_1, x_2) \in X \times X : (x_1, x_2) = \sigma S(x_1, x_2), \sigma \in ]0, 1[\}$ is bounded:

Let $(y_1, y_2) \in A$. Then, we have $(y_1, y_2) = \sigma S(y_1, y_2)$ for some $0 < \sigma < 1$.

So, we have $y_i = \sigma S_i(y_1, y_2); i = 1, 2$.

Hence, we can write

$$
\begin{aligned}
\|y_i\|_\infty \quad \leq \quad & \sigma \max_{i = 1,2} \left( \left[ N_{1_i} + N_{2_i} + N_{3_i} + N_{4_i} \right] \left[ \left( 2 + \frac{|\theta_i^*||\theta_i^{q_i + 1}}{\Gamma(q_i + 2)} \right) \frac{1}{\Gamma(\alpha_i + \beta_i + 1)} \right. \right. \\
& + \left( \frac{2}{(\alpha_i - 1)} + |\nu_{1_i}| \right) \frac{1}{\Gamma(\alpha_i + \beta_i)} + \frac{\theta^{\alpha_i + \beta_i + q_i}}{\Gamma(\alpha_i + \beta_i + q_i + 1)} \\
& \left. + \left( \frac{2}{(\alpha_i - 1)\Gamma(\alpha_i + 1)} + \frac{\alpha_i}{(\alpha_i - 1)\Gamma(\alpha_i + 2)} + |\nu_{2_i}| \right) \frac{1}{\Gamma(\beta_i + 1)} \right] \\
& + \left( \frac{1 + \alpha_i}{(\alpha_i - 1)\Gamma(\alpha_i + 1)} + \frac{1}{(\alpha_i - 1)\Gamma(\alpha_i + 2)} + |\nu_{3_i}| \right) r \chi_i \mu_i \\
& + k_i r \Gamma(1 - \lambda_i) \left[ \left( 2 + \frac{|\theta_i^*||\theta_i^{q_i + 1}}{\Gamma(q_i + 2)} \right) \frac{1}{\Gamma(\alpha_i - \lambda_i + 1)} + \left( \frac{2}{(\alpha_i - 1)} + |\nu_{1_i}| \right) \frac{1}{\Gamma(\alpha_i - \lambda_i)} \right. \\
& \left. \left. + \frac{1}{\Gamma(\alpha_i - \lambda_i + q_i + 1)} \right] \right).
\end{aligned}
$$

It is also evident that

$$
\|D^{\delta_i} y_i\|_\infty
$$

$$
\leq \ \sigma \max_{i=1,2}\left(\left[N_{1_i} + N_{2_i} + N_{3_i} + N_{4_i}\right]\left[\frac{1}{\Gamma(\alpha_i - \delta_i + \beta_i + 1)} + \frac{1}{\Gamma(\alpha_i + \beta_i + 1)\Gamma(2 - \delta_i)}\right.\right.
$$

$$
+ \left(\frac{\Gamma(\alpha_i + 1)}{(\alpha_i - 1)\Gamma(\alpha_i - \delta_i + 1)} + \frac{\Gamma(\alpha_i + 1)}{(\alpha_i - 1)\Gamma(\alpha_i - \delta_i + 2)} + \frac{\alpha_i}{(\alpha_i - 1)(1 + \alpha_i)\Gamma(2 - \delta_i)}\right)\frac{1}{\Gamma(\alpha_i + \beta_i)}
$$

$$
+ \left(\frac{1}{(\alpha_i - 1)\Gamma(\alpha_i + 1)\Gamma(2 - \delta_i)} + \frac{1}{(\alpha_i - 1)\Gamma(\alpha_i - \delta_i + 2)}\right)\frac{1}{\Gamma(\beta_i + 1)}\Bigg]
$$

$$
+ \left(\frac{1}{(\alpha_i - 1)\Gamma(\alpha_i - \delta_i + 1)} + \frac{\alpha_i(\alpha_i + 1)}{(\alpha_i - 1)\Gamma(\alpha_i - \delta_i + 2)} + \frac{1}{(\alpha_i - 1)\Gamma(\alpha_i + 2)\Gamma(2 - \delta_i)}\right)r\chi_i \mu_i
$$

$$
+ k_i r \Gamma(1 - \lambda_i)\left[\frac{1}{\Gamma(\alpha_i - \delta_i - \lambda_i + 1)} + \frac{1}{\Gamma(\alpha_i - \lambda_i + 1)\Gamma(2 - \delta_i)}\right.
$$

$$
\left.\left. + \left(\frac{\Gamma(\alpha_i + 1)}{(\alpha_i - 1)\Gamma(\alpha_i - \delta_i + 1)} + \frac{\Gamma(\alpha_i + 1)}{(\alpha_i - 1)\Gamma(\alpha_i - \delta_i + 2)} + \frac{\alpha_i}{(\alpha_i - 1)(1 + \alpha_i)\Gamma(2 - \delta_i)}\right)\frac{1}{\Gamma(\alpha_i - \lambda_i)}\right]\right).
$$

Using Equations (11) and (12), we state that $\|(y_1, y_2)\|_{X\times X} < \infty$. The set is thus bounded.

As a consequence of Schaefer fixed point theorem, we conclude that $S$ has a fixed point which is a solution of Equation (1). $\quad\square$

## 4. Example

In this section, we present an example to illustrate the application of the first main result. Consider the following system:

$$
\begin{cases}
D^{\frac{3}{2}}(D^{1.98} + \frac{5}{10^4 t^{0.02}})y_1(t) + \frac{|y_1(t) + D^{\frac{1}{2}}y_2(t)|}{40 e^{t+2}(1 + |y_1(t) + D^{\frac{1}{2}}y_2(t)|)} + \frac{\cos y_1(t) + \cos I^{\frac{1}{2}}y_2(t)}{64\pi^2} \\[2mm]
+ \frac{\sin y_1(t)}{444 e^{t^2 + t}} = \frac{t}{5}, \ \ 0 < t < 1, \\[4mm]
D^{1.7}(D^{1.9} + \frac{1}{10^3 t^{\frac{5}{1000}}})y_2(t) + \frac{1}{80 e^t}\left(\frac{|y_1(t)|}{2(1 + |y_1(t)|)} + \frac{\cos D^{0.2}y_2(t)}{e^{t^2 + 1}}\right) + \frac{1}{50}\left(\frac{\sin y_1(t)}{e^{t+2}} + \frac{\sin I^{0.8}y_2(t)}{12(t+1)}\right) \\[2mm]
+ \frac{|y_2(t)|}{100\pi(1 + |y_2(t)|)} = \frac{2t}{3}, \ \ 0 < t < 1, \\[4mm]
y_i(1) = y_i(0), \ \ i = 1, 2, \\[4mm]
y_i'(1) = y_i'(0), \ \ i = 1, 2, \\[4mm]
D^{1.98}y_1(1) + \frac{5}{10^4}y_1(1) = \int_0^{0.1} \frac{\tau}{1000}y_1(\tau)d\tau, \\[2mm]
D^{1.9}y_2(1) + \frac{1}{10^3}y_2(1) = \int_0^{0.4} \frac{5\tau}{1000}y_2(\tau)d\tau, \\[4mm]
I^{0.2}y_1(0.7) = y_1(1), \\[2mm]
I^{0.1}y_2(\frac{1}{2}) = y_2(1).
\end{cases}
$$

We have

$$f_1(t, u_1(t), u_2(t)) = \frac{|u_1(t) + u_2(t)|}{40e^{t+2}(1 + |u_1(t) + u_2(t)|)},$$

$$f_2(t, u_1(t), u_2(t)) = \frac{1}{80e^t} \left( \frac{|u_1(t)|}{2(1 + |u_1(t))|} + \frac{\cos u_2(t)}{e^{t^2+1}} \right),$$

$$g_1(t, u_1(t), u_2(t)) = \frac{\cos u_1(t) + \cos u_2(t)}{64\pi^2},$$

$$g_2(t, u_1(t), u_2(t)) = \frac{1}{50} \left( \frac{\sin u_1(t)}{e^{t+2}} + \frac{\sin u_2(t)}{12(t+1)} \right),$$

$$h_1(t, u(t)) = \frac{\sin u(t)}{444e^{t^2+t}},$$

$$h_2(t, u(t)) = \frac{|u(t)|}{100\pi(1 + |u(t))|)},$$

and

$$\alpha_1 = 1.98, \beta_1 = \frac{3}{2}, \rho_1 = \frac{1}{2}, \delta_1 = \frac{1}{2}, \mu_1 = 0.1, k_1 = \frac{5}{10^4}, \lambda_1 = 0.02, \theta_1 = 0.7, q_1 = 0.2, \chi_1 = \frac{1}{1000},$$

$$\alpha_2 = 1.9, \beta_2 = 1.7, \rho_2 = 0.8, \delta_2 = 0.2, \mu_2 = 0.4, k_2 = \frac{1}{10^3}, \lambda_2 = \frac{5}{1000}, \theta_2 = \frac{1}{2}, q_2 = \frac{1}{10}, \chi_2 = \frac{5}{1000}.$$

We can also consider that

$$\Delta_{f_1} = \frac{1}{40e^2}, \Delta_{g_1} = \frac{1}{64\pi^2}, r_1 = \frac{1}{444}, \Delta_{f_2} = \frac{1}{160}, \Delta_{g_2} = \frac{1}{50e^2}, r_2 = \frac{1}{100\pi}.$$

So, we obtain:

$$\varepsilon_1 1 = 0.2205, \quad \varepsilon_1 2 = 0.0253, \qquad \varepsilon_2 1 = 0.2191, \quad \varepsilon_2 2 = 0.04,$$
$$\varepsilon_1 = max\{\varepsilon_1 1, \varepsilon_1 2\} = 0.2205, \qquad \varepsilon_2 = max\{\varepsilon_2 1, \varepsilon_2 2\} = 0.2191.$$

By Theorem 1, we confirm that the above example has a unique solution.

## 5. Conclusions

The fundamental objective of this work was to introduce the new fractional coupled system of Lane–Emden type given by Equation (1). By using the well known Banach fixed point theorem combined with the integral inequality theory, a first main result on the existence of a unique solution has been proved. Then, by application of Scahefer fixed point theorem, another main result, that studied the existence of at least one solution for Equation (1), has been discussed. Finally, we have discussed an illustrative example.

The reader can see that in this paper, the question of stability of solutions in the sense of Ulam Hyers has not been considered. This type of stability and some others will be investigated in the future paper. This work is in progress.

**Author Contributions:** Z.D. and S.E.F. analyzed the problem and Y.G. suggested mathematical modeling. F.A. authenticated the results and supervised the research overall. This manuscript has been written by S.E.F., she also provided answers to the reviewers comments. All authors have read and agreed to the published version of the manuscript.

**Funding:** No funding available from any agency.

**Acknowledgments:** The authors are highly thankful to NTU Singapore, UOH KSA, GCU, Lahore, Pak and LPAM, UMAB, Algeria for providing high quality research environment.

**Conflicts of Interest:** The authors have no competing interests with anyone or with any institution.

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
