# Peer review of "Fractional Singular Differential Systems of Lane–Emden Type: Existence and Uniqueness of Solutions"

_axioms, doi:10.3390/axioms9030095_

Round 1
Reviewer 1 Report
- A poor introduction is given and is similar to reference [13].
- Improve the presentation of some equations (2.4), (3.3),...
- The manuscript has some mistakes for example: In the page 3, below equation (1.1) change \rho to \rho_i, in the definition 1, change 0<\alpha<1 to \alpha>0, in the definition 2, change the function space,...
- The authors do not show the stability analysis in the manuscript.
- Add a conclusions section.
Author Response
Responses to the Comments of the Reviewer # 1
The authors are highly obliged about the comments/guidelines of the reviewer # 1. We strongly agree with the reviewer and fully accept the comments / guidelines to follow. We have tried our best to respond and address all comments and guidelines in the manuscript accordingly. We own that the respected reviewer has full command on the subject and manuscript analysis.
Kindly accept our responses as we are followers of the great mathematicians like you and surely we are beginners only in high mathematics.
Reviewer 01:
- Comment 1: A poor introduction is given and is similar to reference [13].
- Response 1: We have improved the introduction and review of the literature, as well as, we have added references that support more the content of the ideas of the introduction. PLEASE, see the revised version of our paper.
- Comment 2: Improve the presentation of some equations (2.4), (3.3).
- Response 2 :We have improved the presentation of equations, for example, see attachment
- Comment 3: The manuscript has some mistakes for example: In the page 3, below equation (1.1) change \rho to \rho_i, in the definition 1, change 0<\alpha<1 to \alpha>0, in the definition 2, change the function space,...
- Response 3: The required errors are corrected. For the function space, we think that the C^n space is the space for which we can speak about Caputo derivative. The space is considered as a sufficient condition to the existence of the derivative of Caputo. Here, we kindly ask the referee to propose for us another function space if there exists.
Thank you in advance.
- Comment 4: The authors do not show the stability analysis in the manuscript.
- Response 4: The present paper deals with the existence and uniqueness as well as the existence of at least one solution. For the question of stability, we have not investigated it in this paper, it will be considered in the future paper. We invite the referee to see the conclusion of the revised version where we have indicated that the stability will be studied in the future paper. Thank you.
- Comment 5: Add a conclusions section.
- Response 5: Also, we added the conclusion part. Thank you.
Thank you so much, for reviewing our manuscript again.

Reviewer 2 Report
The paper is reviewed very carefully and is a good contribution. Hence I recommend this paper for publication after the following minor revisions.
Please check the manuscript carefully for typos and grammar errors and correct them.
Check the format of the journal and made all the references according to the journal style.
Highlights although they are correct need to be extended, in order to mention the main results of the investigation.
Some minor fingerprints mistakes appear in the manuscript, a detailed revision of these minor spelling errors is needed in order to improve this work.
Authors should include following references for better presentation of paper
- The paper needs to improve the introduction involving the recent advancements in fractional calculus and its applications see:
A numerical solution for a variable-order reaction-diffusion model by using fractional derivatives with non-local and non-singular kernel. Physica A: Statistical Mechanics and its Applications. (2018).
Decolonisation of fractional calculus rules: Breaking commutativity and associativity to capture more natural phenomena. The European Physical Journal Plus. 133, (2018), 1-23.
Non validity of index law in fractional calculus: A fractional differential operator with Markovian and non-Markovian properties. Physica A: Statistical Mechanics and its Applications. 505, (2018), 688-706.
I want to read speedly the last version of paper before publishing if it possible for you.
Author Response
Responses to the Comments of the Reviewer # 2
The authors are highly obliged about the comments/guidelines of the reviewer # 2. We strongly agree with the reviewer and fully accept the comments / guidelines to follow. We have tried our best to respond and address all comments and guidelines in the manuscript accordingly. We own that the respected reviewer has full command on the subject and manuscript analysis.
Kindly accept our responses as we are followers of the great mathematicians like you and surely we are beginners only in high mathematics.
Reviewer 02:
- Comment 1: Please check the manuscript carefully for typos and grammar errors and correct them.
- Response 1: The manuscript has been carefully checked with spelling and grammatical errors checked and corrected.
- Comment 2:Check the format of the journal and made all the references according to the journal style.
- Response 2: References are organized according to journal style, (and we will also correct any other thing that can exist in the last version in the case of acceptance of this paper). Many thanks.
- Comment 3: Highlights although they are correct need to be extended, in order to mention the main results of the investigation.
- Response 3: We think also that this is an important remark. We have extended the highlights throughout the paper, please see the revised version and in particular in the Introduction Section. Thank you.
- Comment4: Some minor fingerprints mistakes appear in the manuscript, a detailed revision of these minor spelling errors is needed in order to improve this work.
- Response 4: The errors in the manuscript are corrected, and spelling errors are revised. If there exist some others, we will correct them in the last version of the journal. Many thanks.
- Comment 5: Authors should include following references for better presentation of paper. The paper needs to improve the introduction involving the recent advancements in fractional calculus and its applications see:
- A numerical solution for a variable-order reaction-diffusion model by using fractional derivatives with non-local and non-singular kernel. Physica A: Statistical Mechanics and its Applications. (2018).
- Decolonization of fractional calculus rules: Breaking commutatively and associativity to capture more natural phenomena. The European Physical Journal Plus. 133, (2018), 1-23.
- Non validity of index law in fractional calculus: A fractional differential operator with Markovian and non-Markovian properties. Physica A: Statistical Mechanics and its Applications. 505, (2018), 688-706.
- Response 5: The introduction is improved by mentioning the proposed references.
Thank you so much, for reviewing our manuscript again.

Reviewer 3 Report
Dear editor,
Please see the attachment.
Best wishes,

Author Response
Responses to the Comments of the Reviewer # 3
The authors are highly obliged about the comments/guidelines of the reviewer # 3. We strongly agree with the reviewer and fully accept the comments / guidelines to follow. We have tried our best to respond and address all comments and guidelines in the manuscript accordingly. We own that the respected reviewer has full command on the subject and manuscript analysis.
Kindly accept our responses as we are followers of the great mathematicians like you and surely we are beginners only in high mathematics.
Reviewer # 3:
- Comment 1: Lemma 3 is important. So, its proof is not clear. It should be justified.
- Response 1: We have justified Lemma 3, and we kindly propose to the reviewer to see the last version of this paper. Thank you for this remark.
- Comment 2: What is the definition of ? on Page 6?
- Response 2: We mean ?i. Please see problem (1.1). It is corrected. Thank you.
- Comment 3: Hypotheses T1 is overlap with the introduction about ℎ?.
- Response 3: These overlaps are eliminated, by removing continuity while introducing the problem. Please see the T1 of this revised version. Thank you.
- Comment 4: In the hypotheses T3, why should ? be in ?4?
- Response 4: We have corrected it: ?2.
- Comment 5: In the hypotheses T4, what is the definition of ?4??
- Response 5: It is a quantity that is well defined in the page 07. Thank you.
- Comment 6: How to get (3.6)? It lacks the proof. Similarly, (3.7). Many important equations are lack their corresponding proof.
- Response 6: We have clarified in detail now, that how some results are obtained, and some others are given in summary form to avoid repetition. For example, Proof (3.6): you can see below: see attachment
- Comment 7: There are many punctuation, format and spelling errors in the paper.
- Response 7: Several errors are corrected in the paper. If there are still some others, we will do our best in the last version in the case of acceptance.
Thank you so much, for reviewing our manuscript again.

Round 2
Reviewer 1 Report
The manuscript improved a lot and my observations were attended by the authors.
My observations are as follows:
- In the equation (2.1) correct the signs at the end of the equation.
- After equation (3.2), start with a capital letter.
- In the hypothesis (T3) change "M1i, M2i, ..." for "N1i, N2i,...
Author Response
Responses to the Comments of the Reviewer # 1 Round 2
The authors are highly obliged about the comments/guidelines highlighted by the reviewer # 1. We strongly agree with the reviewer and fully accept the comments / guidelines to follow. We have tried our best to respond and address all comments and guidelines in the manuscript accordingly. We own that the respected reviewer has full command on the subject and manuscript analysis.
Kindly accept our responses as we have followed guidelines accordingly.
Reviewer 01:
- Comment 1: In the equation (2.1) correct the signs at the end of the equation.
- Response 1: Its now corrected accordingly. Thanks a lot.
- Comment 2: After equation (3.2), start with a capital letter.
- Response 2: The error has been corrected accordingly. Thanks a lot.
- Comment 3: In the hypothesis (T3) change "M1i, M2i, ..." for "N1i, N2i, ...
- Response 3: Also, the errors mentioned by the reviewer #1, have been corrected accordingly. Thanks a lot once again.

Reviewer 3 Report
I suggest this manuscript can be published after the following minor revisions:
Some format of mathematical formulas should be checked again. They should be ordered, neat and standard.
Author Response
Responses to the Comments of the Reviewer # 3 Round 2
The authors are highly obliged about the comments/guidelines highlighted by the reviewer # 3. We strongly agree with the reviewer and fully accept the comments / guidelines to follow. We have tried our best to respond and address all comments and guidelines in the manuscript accordingly. We own that the respected reviewer has full command on the subject and on manuscript analysis.
Reviewer 03:
- Comment 1: Some format of mathematical formulas should be checked again. They should be ordered, neat and standard.
Kindly accept our responses as we have followed the instruction accordingly and have corrected / formatted almost all formulas. If some of them still ill formatted kindly ignore for the time being as on acceptance (if) then for final version/ Galley proof, we must do accordingly.
